# Relationship Between Retinal Microcirculation and Renal Function in Patients with Diabetes and Chronic Kidney Disease by Laser Speckle Flowgraphy

**DOI:** 10.3390/life13020424

**Published:** 2023-02-02

**Authors:** Takeshi Iwase, Yoshitaka Ueno, Ryo Tomita, Hiroko Terasaki

**Affiliations:** 1Department of Ophthalmology, Akita University Graduate School of Medicine, Akita 010-8543, Japan; 2Department of Ophthalmology, Nagoya University Graduate School of Medicine, Nagoya 466-8550, Japan

**Keywords:** chronic kidney disease, laser speckle flowgraphy, diabetic retinopathy, blood flow, diabetic nephropathy

## Abstract

This study investigated the effect of renal dysfunction categorized by the stage of chronic kidney disease (CKD) on the retinal microcirculation assessed by laser speckle flowgraphy (LSFG) and retinal artery caliber measured by adaptive optics imaging in diabetic patients particularly the early stage of retinopathy and nephropathy. We divided the patients with diabetes into three groups based on the CKD stage (non-CKD (*n* = 54); CKD stage 1 + 2 (*n* = 20); CKD stage 3 (*n* = 41)). The mean blur rate (MBR) of the stage 3 CKD group was significantly lower than that of the no-CKD group (*p* < 0.015). The total retinal flow index (TRFI) in the stage 3 CKD group was significantly lower than that of the no-CKD group (*p* < 0.002). Multiple regression analysis demonstrated that CKD stage was independently associated with MBR (β = −0.257, *p* = 0.031) and TRFI (β = −0.316, *p* = 0.015). No significant differences were observed in external diameter, lumen diameter, wall thickness, and wall to lumen ratio among the groups. These results indicated that the ONH MBR and TRFI as assessed by LSFG decreases in diabetic patients with stage 3 CKD, but the arterial diameter measured by adaptive optics imaging does not change, suggesting that impaired renal function may be associated with decreased retinal blood flow in early-stage diabetic retinopathy.

## 1. Introduction

The incidence and morbidity of diabetes mellitus have increased in the world. Diabetic retinopathy (DR) is one of the main causes of blindness, and about 40% of patient with diabetes develop diabetic nephropathy (DN), causing end-stage renal disease in developed countries [1]. The retinal and the kidney-related complications of DM both result from microvascular damage in these organs. Carlson reported that the basement membrane in the retinal and glomerular capillary vessels thickens in end-stage diabetic retinopathy and nephropathy [2], and He et al. reported that proliferative diabetic retinopathy is a highly specific indicator for the diagnosis of DN [3]. Those findings show the anatomical and clinical association between DR and DN in the end stage. Although approximately one-third of patients who do not have either albuminuria or retinopathy suffer from decreased kidney function [4], few studies have evaluated the relationship between retinal morphology and renal function in patients with the early stage of retinopathy and nephropathy.

There have been some reports that changes in the caliber of retinal vessels are related to chronic kidney disease (CKD) using direct ophthalmoscopic examinations [5,6]. However, the resolution of fundus photographs is not sufficient for identifying retinal vessel walls and measuring their thickness. Adaptive optics technology noninvasively corrects optical wavefront aberrations, enabling high-definition images of photoreceptor cells and retinal vessel walls that cannot be detected in color fundus photographs [7,8].

Laser speckle flowgraphy (LSFG) (Softcare Co., Ltd., Fukutsu, Japan) can evaluate the ocular blood flow [9,10,11]. The values correlate well with those of the hydrogen gas clearance method and with actual blood flow values measured by the microsphere method [12,13], meaning that variables measured by LSFG are comparable between individuals. Accordingly, LSFG would be appropriate for evaluating ocular blood flow in large clinical trials.

Although there have been studies of retinal and choroidal hemodynamics in diabetic patients with LSFG, it is not known whether impaired retinal microcirculation is associated with renal dysfunction in diabetic patients. Recently, it was reported that CKD is significantly associated with ocular disorders such as cataract and DR [14], indicating that renal dysfunction may have some effect on ocular diseases. However, there are few reports that examine the relationship between retinal microcirculation or actual vessel caliber and renal impairment in diabetes.

Therefore, the purpose of this study was to examine the effect of renal dysfunction categorized by the stage of CKD on the retinal microcirculation assessed by LSFG and retinal artery caliber measured by AO imaging in diabetic patients, especially in the early stage of retinopathy and nephropathy.

## 2. Materials and Methods

### 2.1. Study Design and Ethics

The study was conducted retrospectively and cross-sectionally at the Nagoya University Hospital, and the procedures used were approved by the Ethics Committee of the Nagoya University Hospital (2016-0028-8183) (Nagoya, Japan). This study conformed to the tenets of the Declaration of Helsinki. Patient consent was waived due to the retrospective nature of the study.

### 2.2. Participants

This study enrolled 115 consecutive Japanese diabetic patients from April 2016 to March 2018. Diabetes was diagnosed in patients based on the criteria of the American Diabetes Association [15]. Hypertension was diagnosed if patients’ systolic blood pressure (BP) exceeded 140 and diastolic BP exceeded 90 mmHg or if they took antihypertensive drugs [16]. Dyslipidemia was diagnosed in subjects with serum LDL cholesterol levels ≥ 140 mg/dL and/or HDL cholesterol levels < 40 mg/dL and/or triglyceride levels ≥ 150 mg/dL and a history of cholesterol-lowering treatment [17].

Urinary albumin excretion is expressed as an albumin/creatinine ratio (ACR) (mg/g creatinine). DN was staged based on analysis of spot urine. That is, stage 1 DN (normal albuminuria), ACR < 30 mg/g creatinine; stage 2 DN (trace albuminuria), 30 < ACR < 300 mg/g creatinine; stage 3 DN (giant albuminuria), ACR > 300 mg/g creatinine (or dipstick urine analysis is 2+, 3+, 4+) [18]. If the ACR of the subjects was not measured, dipstick urinalysis was used as reference, i.e., normoalbuminuria, dipstick urine protein negative or trace; microalbuminuria, dipstick urine protein 1+; macroalbuminuria, dipstick urine protein 2+ or more [19].

We measured the serum creatinine level within 4 h after fasting venous blood collection. We also evaluated renal function based on the estimated glomerular filtration rate (eGFR) calculated using published formulas [20]. We classified CKD stages according to the National Kidney Foundation Disease Outcomes Quality Initiative clinical practice guidelines [21] and determined that if there was no trace albuminuria and the eGFR was greater than 60 mL/min/1.73 m^2^, there was no CKD.

In the current study, we enrolled diabetic patients with no or minimal diabetic retinopathy and no albuminuria or microalbuminuria. We excluded patients who had macroalbuminuria or proteinuria, poorly controlled diabetes (HbA1c, 10.0%), uncontrolled hypertension (BP, 160/100 mmHg), severe anemia (hemoglobin, 10.0 g/dL), or other kidney diseases and those who underwent hemodialysis.

For ophthalmologic evaluation, slit-lamp examinations and indirect ophthalmoscopy were used. After the pupils were dilated, an ophthalmologist assessed the retinopathy. We excluded patients with poor visual acuity (<20/25), high intraocular pressure (IOP) (>20 mmHg), moderate-to-severe nonproliferative and proliferative diabetic retinopathy, any macular abnormalities, topical anti-glaucoma treatment, anti-VEGF therapy or steroid at last 1 year before the examinations, or axial length (AL) > 26.5 mm [22]. This study included one eye of each patient.

### 2.3. Laser Speckle Flowgraphy

LSFG-NAVI was used to evaluate the relative ocular blood flow using mean blur rate (MBR) [23,24,25]. The device consists of a fundus camera equipped with an 830 nm diode laser and a charge-coupled camera (750 horizontal × 360 vertical pixels). When the laser is irradiated, the scattered light from the irradiated tissue interferes and a speckle pattern appears. A circle was set surrounding the optic nerve head (ONH) to examine the ONH blood flow (Figure 1). Using LSFG, we measured the ONH blood flow twice for each time point.

We evaluated the relative flow volume (RFV) index using the LSFG-NAVI. The retinal vascular MBR always contains a background of choroidal blood flow. However, the effect of choroidal blood flow can be reduced by manually selecting a region of interest centered on a region where retinal vessels are dominant and subtracting the background choroidal blood flow from the overall MBR. The resulting LSFG value is the RFV.

A total RFV index of the major retinal vessels around the ONH can be semi-automatically calculated as TRFI from the sum of retinal arteries and retinal veins. Rectangular regions were set around each major blood vessel around the ONH, and arterial and venous TRFIs were calculated after semi-automatic identification of arteries and veins (Figure 1). If there was a mistake in the placement, we corrected the rectangle manually.

### 2.4. Adaptive Optics Imaging

An adaptive optics (AO) camera (rtx1, Imagine Eyes, Orsay, France) was used to obtain images of the retinal structures for clinical research [26]. The AO fundus camera produces images of the retina with time by recording the scattered light from a focused beam. The AO fundus camera has three main components integrated into the imaging system to correct for ocular aberrations: a wavefront sensor, a correction element, and a control system to improve image quality [27]. The measured refraction is integrated into the camera, and a live display of the AO-corrected fundus image allows the user to adjust brightness, contrast, and focus.

In the present study, arterioles in zone B, 0.5 to 1 disc diameter away from the optic disc edge, were photographed as the vascular measurement area [28], and the vascular measurements were analyzed using the software to detect arteries (Figure 1). The external diameter (ED), lumen diameter (LD), wall thickness (WT), and wall to lumen ratio (WLR) values for the retinal artery were obtained by two retina specialists (T.I. and Y.U.) as the vascular measurements.

### 2.5. Statistical Analyses

The value of each parameter was expressed as the mean ± standard deviation. We used one-way ANOVA for continuous variables and the x^2^ test for categorical variables to compare between groups. After one-way ANOVA, post hoc comparisons were performed using the Tukey–Kramer procedure. We used Spearman’s rank correlation coefficient tests to evaluate the correlation coefficients between the variables and used multiple stepwise regression analysis to investigate the association between blood flow parameters and other variables. IBM SPSS Statistics for Windows, v.23 (IBM Corp., Armonk, NY, USA), was used for statistical analyses. A probability (*p*) value < 0.05 was set as the significance level.

## 3. Results

### 3.1. Demographics of Subjects

The diabetic patients were divided into three groups based on the CKD stage (non-CKD, *n* = 54; CKD stage 1 + 2, *n* = 20; CKD stage 3, *n* = 41). No patient had stage 4 or 5 CKD. Table 1 shows demographic data. There were no significant differences in gender, HbA1c, duration of diabetes, SBP, DBP, IOP, HR, Hb, total cholesterol, triglyceride, or LDL among the groups, while there were significant differences in age (*p* = 0.001), serum creatinine (*p* < 0.001), and eGFR (*p* < 0.001) among the groups.

### 3.2. Comparison of Ocular Blood Flow among the Groups

In the no-CKD group, the ONH MBR in the vessel, the ONH MBR in the tissue, the choroidal MBR, and the TRFI were 42.85 ± 7.48, 11.33 ± 1.82, 8.50 ± 2.77, and 2768.53 ± 752.08 (arbitrary unit, AU), respectively (Figure 2). In the stage 1 + 2 CKD group, those were 40.73 ± 6.82, 11.34 ± 2.31, 8.61 ± 3.22, and 2533.62 ± 720.37 AU, respectively. In the stage 3 CKD group, those were 38.51 ± 7.47, 11.29 ± 2.17, 8.39 ± 3.54, and 2232.18 ± 668.98 AU, respectively. The ONH MBR in the vessel area of the stage 3 CKD group was significantly lower than that of the no-CKD group (*p* < 0.015). The TRFI in the stage 3 CKD group was significantly lower than that of the no-CKD group (*p* < 0.002). There were no significant differences in the ONH MBR in the tissue area and the choroidal MBR among the groups.

### 3.3. Comparison of Retinal Vascular Measurements

Clear vascular images were successfully acquired for 36 of 54 eyes in the no-CKD group, 15 of 20 eyes in the stage 1 + 2 CKD group, and 28 of 41 eyes in the stage 3 CKD group. Clear images could not be obtained for the remaining eyes because of poor eye fixation. Vascular measurements for all subjects are shown in Table 2. No significant differences were observed in external diameter, lumen diameter, wall thickness, and wall to lumen ratio among the groups.

### 3.4. Correlation of Ocular Blood Flow with Other Parameters

The results of single linear regression analysis are displayed in Table 3. The ONH MBR in the vessel correlated with age (r = −0.450, *p* < 0.001), duration of DM (r = −0.238, *p* = 0.016), eGFR (r = 0.266, *p* < 0.004), CKD stage (r = −0.250, *p* < 0.007), and DBP (r = 0.223, *p* = 0.017). The TRFI correlated with age (r = −0.556, *p* < 0.001), duration of DM (r = −0.299, *p* = 0.003), eGFR (r = 0.331, *p* < 0.001), and CKD stage (r = −0.313, *p* = 0.001).

Multiple regression analysis showed that CKD stage was independently associated with ONH MBR-vessel and TRFI (Table 4 and Table 5).

## 4. Discussion

Our results showed that ONH MBR-vessel and TRFI were significantly reduced in the stage 3 CKD group compared with the no-CKD group. In vascular measurements of the retinal artery by AO imaging, no significant differences were observed in external diameter, lumen diameter, wall thickness, and wall to lumen ratio among the groups. Multiple regression analysis showed that CKD stage was a common independent factor associated with ONH MBR-vessel and TRFI in subjects.

In this study, retinal blood flow velocity and volume was decreased in the stage 3 CKD group, while the vascular parameters in retinal artery determined by AO imaging were not significantly different among the groups. There have been several reports on the association between CKD and retinal vessel [29,30,31,32], but no certain view has been reached.

The Cardiovascular Health Study [29], the Beaver Dam CKD Study [30], and McGowan [31] reported no association between changes in vessel caliber and decreased renal function. On the other hand, the Atherosclerosis Risk in Communities Study reported narrowing of retinal arterioles with worsening creatinine over a 6-year course [32]. More recently, Nagaoka et al. reported narrower arteriolar diameters in patients in the stage 3 CKD group than in patients in the no-CKD group using laser Doppler velocimetry in early-stage diabetic retinopathy [33].

The measurement of vessel diameter by a fundus camera reflects the blood column surrounded by the plasma edge stream and may underestimate the actual inner diameter [34], and the same principle may apply to laser Doppler velocimetry imaging [35]. On the other hand, AO camera imaging is thought to be able to measure the actual inner and outer diameter of retinal vessels. However, no significant differences were observed in the measurement of vessel diameter among the groups using AO imaging in the study.

Since it has been reported that retinal and glomerular capillaries in end-stage diabetic retinopathy and nephropathy have a thickened basement membrane [2], it is thought that the retinal arterioles become narrower when renal function declines to the point that dialysis is eventually performed. However, because the patients in this study had CKD only up to stage 3, the retinal vessels may not have been significantly different in their early atherosclerotic morphology. This study suggests that, in patients with CKD, changes in blood flow may occur first rather than changes in vessel diameter in the early stages of diabetic retinopathy.

Shiba et al. used LSFG to show that, in diabetic patients without retinopathy, BOS (pulse waveform analysis tuned to the heartbeat, which manifests the variation in MBR between systolic and diastolic) in the ONH was lower than in normal eyes, indicating greater stress on the vessel wall during blood pressure changes [36]. Nagaoka et al. also reported that decreased retinal blood flow in patients with early or nondiabetic retinopathy using laser Doppler velocimetry [37]. These reports indicate that hemodynamic changes in the eye occur even in early diabetic retinopathy.

Although DR and the preclinical morphologic changes of DN are associated closely [38], about one-third of diabetic patients have decreased kidney function without either albuminuria or retinopathy [4].

Nagaoka et al. reported that, using laser Doppler velocimetry, the retinal blood flow in the retinal arterioles decreased in patients with stage 3 CKD compared to patients without CKD [33], suggesting that impaired renal function would be associated with decreased retinal microcirculation in early-phase diabetic retinopathy. TRFI (indicating retinal blood flow volume) was reduced in patients with stage 3 CKD in our study, which corroborates Nagaoka’s report. Those results suggest that the decrease in retinal blood flow is caused by the reduction of retinal blood flow velocity, because the retinal vessel parameters were not different. The AO camera can measure retinal vessels accurately, as aforementioned, and it is more likely that the reduction of retinal blood volume is caused by the decrease of blood flow velocity.

In this study, blood flow was lower in the stage 3 group than in the no-CKD group due to a decrease in blood flow velocity, although there was no change in arterial diameter. There are several considerable reasons for this. First, endothelial function in retinal vessels should be reduced. CKD has been associated with mild inflammation, platelet activation, and decreased vascular endothelial cell function [39]. In addition, von Willebrand factor (VWF) levels, a marker of impaired vascular endothelial cell function, have been reported to be higher in diabetic patients than in normal subjects, and even higher in the presence of retinopathy and nephropathy [40]. Feng et al. reported an association between high VWF levels and reduced ocular blood flow in early retinopathy in type 1 diabetic patients [41], indicating the relationship between the measured endothelial cell function and reduced ocular blood flow. Although we did not measure vascular endothelial cell function in the present study, these findings suggest that reduced endothelial cell function in diabetic patients with stage 3 CKD results in reduced blood flow velocity.

Second, blood viscosity is increased in patients with CKD. According to the Hagen–Poiseuille law, increased blood viscosity results in reduced blood flow and increased vascular resistance in the renal vascular beds [42]. Sugimori et al. reported that increased blood viscosity is associated with reduced renal function and increased urinary albumin excretion [43]. In the current study, decreased renal function may have increased blood viscosity and decreased retinal circulation.

In addition, increased blood viscosity decreases blood flow and oxygen supply to retinal ganglion cells [44] and is a risk factor for diabetic retinopathy [45], retinal artery and vein occlusion [46], and glaucoma [47]. Clinical study has reported that CKD is associated with a high prevalence of diabetic retinopathy [48], and Wong et al. further showed an association between vision-threatening retinopathy and CKD [49]. Although the present study was conducted in patients with early-stage retinopathy, ophthalmologists need to carefully follow up the patients because ocular blood flow is reduced in the stage 3 CKD group and retinal ischemia may progress to more severe retinopathy or other ocular diseases in the future.

There are several limitations in the present study. First, because the design of the study was cross sectional with a small patient population, a prospective study is required to detect the interaction of the pathogenesis between retinopathy and nephropathy in diabetes with a larger patient population. Second, the average age of the patients in the stage 3 CKD group was higher than that of the patients in the other CKD groups. Because we previously reported that ONH MBR decrease with aging [50], it is also possible that higher age affects the MBR. However, since the prevalence of CKD is also higher with aging [51], this may not have influenced the result negatively. Third, although two retinal specialists confirmed the parameters measured by AO camera imaging such as the WLR, we measured those at one site per eye, but we believe that more-accurate results can be obtained by measuring at multiple sites per eye. Fourth, only patients with early-stage diabetic retinopathy and nephropathy were included, the relationship between retinal hemodynamics or vessel diameter and renal function in patients with end-stage diabetic microvascular complications is still unclear. Further study is needed to obtain new knowledge pertaining to the progressive stages.

In conclusion, our results showed that the ONH MBR-vessel and TRFI by LSFG decreases in patients with diabetes with stage 3 CKD, but the arterial diameter measured by AO camera imaging does not change, indicating that impaired renal function may be associated with decreased retinal blood flow in early-stage diabetic retinopathy. LSFG should be a useful method to detect early changes in retinal microcirculation effected by renal dysfunction.

## Figures and Tables

**Figure 1 life-13-00424-f001:**
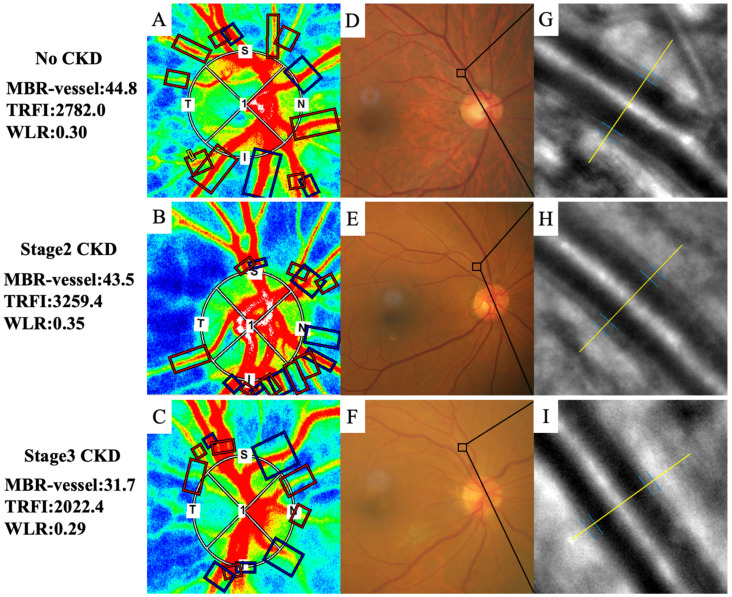
Representative composite color maps of the MBRs on the ONH as measured by LSFG and adaptive optics imaging of the retinal artery in eyes with no CKD, stage 2 CKD, and stage 3 CKD. (**A**–**C**) A circle was set around the ONH when measuring the ONH MBR. To measure the TRFI of the major retinal arteries and veins surrounding the ONH, a rectangle was set at all vessels surrounding the ONH. MBR-vessel and TRFI were lower in patients with stage 3 CKD (**C**) than that in patients with other stages (**A**,**B**). (**D**–**F**) Fundus photographs in eyes with no CKD, stage 2 CKD, and stage 3 CKD. (**G**–**I**) Adaptive optics images of the area outlined in black in panel (**D**–**F**). Wall to lumen ratio (WLR) was 0.30 in no CKD, 0.35 in stage 2 CKD, and 0.29 in stage 3 CKD.

**Figure 2 life-13-00424-f002:**
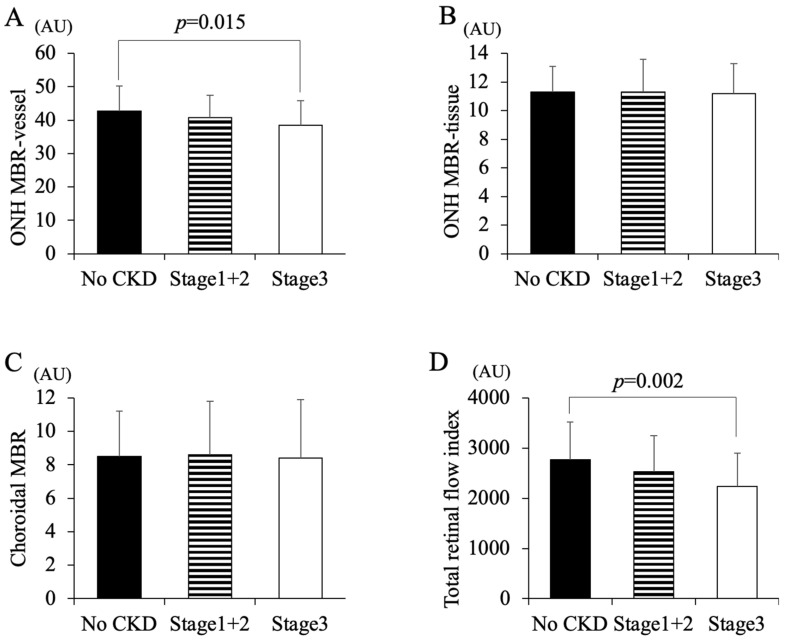
Differences in MBR and TRFI among the no-CKD, stage 1 + 2 CKD, and stage 3 CKD groups. The ONH MBR-vessel in the stage 3 CKD group was significantly lower than that in the no-CKD group (**A**) (*p* = 0.015). The MBR in the tissue and choroid was not significantly different among the groups (**B**,**C**). The TRFI in the stage 3 CKD group was significantly lower than that in the no-CKD group (**D**) (*p* = 0.002).

**Table 1 life-13-00424-t001:** Clinical characteristics of subjects.

Characteristics		Stage of CKD		
	No CKD	Stage 1 + 2	Stage 3	*p*-Value
*n*	54	20	41	-
Age (years)	60.9 ± 12.4	58.05 ± 13.0	69.3 ± 8.4	0.001
Gender (male:female)	25:29	15:5	28:13	0.099
HbA1c (%)	7.7 ± 1.3	7.5 ± 0.9	7.1 ± 1.0	0.065
Duration of diabetes (years)	14.0 ± 11.3	13.2 ± 11.1	17.0 ± 8.9	0.278
Insulin: oral hypoglycemic agents	28:26:00	10:10	12:29	0.072
Serum creatinine, mg/dL	0.67 ± 0.14	0.80 ± 0.16	1.02 ± 0.20	<0.001
eGFR, mL/min/1.73 m^2^	81.4 ± 14.8	75.5 ± 14.8	51.9 ± 7.2	<0.001
Systolic blood pressure (mmHg)	133.4 ± 25.5	137.0 ± 20.2	130.9 ± 19.7	0.614
Diastolic blood pressure (mmHg)	78.6 ± 25.6	81.4 ± 12.9	76.0 ± 10.6	0.249
Intraocular pressure (mmHg)	14.5 ± 3.4	14.9 ± 3.5	14.4 ± 2.6	0.898
Ocular perfusion pressure (mmHg)	50.1 ± 10.1	52.0 ± 9.2	48.4 ± 8.3	0.371
Heart rate, bpm	80.6 ± 12.6	85.2 ± 13.1	76.6 ± 10.1	0.028
Hemoglobin, %	13.5 ± 1.7	14.3 ± 1.8	13.1 ± 1.7	0.044
Total cholesterol, mg/dL	187 ± 46.6	166.7 ± 24.9	174.5 ± 36.7	0.213
Triglycerides, mg/dL	142.2 ± 123.3	151.1 ± 83.7	128.6 ± 73.4	0.695
LDL, mg/dL	105.9 ± 36.9	95.2 ± 19.2	96.0 ± 27.0	0.255
Hypertension, *n* (%)	19 (35)	9 (45)	28 (68)	0.078
Dyslipidemia, *n* (%)	24 (44)	8 (40)	27 (65)	0.063

**Table 2 life-13-00424-t002:** Vessel parameters of subjects obtained using an adaptive optics camera.

Parameter		Stage of CKD		
	No CKD	Stage 1 + 2	Stage 3	*p*-Value
*n*	36	15	28	-
External diameter	119.26 ± 19.72	124.82 ± 11.88	122.44 ± 16.13	0.476
Lumen diameter	90.87 ± 15.84	94.51 ± 9.83	92.60 ±13.57	0.618
Wall	14.43 ± 2.66	14.49 ± 1.64	14.66 ± 2.61	0.938
Wall to lumen ratio	0.31 ± 0.04	0.32 ± 0.04	0.32 ± 0.05	0.733

**Table 3 life-13-00424-t003:** Result of Spearman’s rank correlation coefficient between the MBR or the TRFI and clinical parameters.

Parameter	Age	Duration of DM	HbA1c	T-chol	LDL	TG	eGFR	CKD Stage	SBP	DBP	OPP	HT	DL	WLR
MBR-vessel	−0.450 **	−0.238 *	0.074	−0.064	0.060	0.081	0.266 **	−0.250 **	0.160	0.223 *	0.123	−0.071	−0.025	−0.051
TRFI	−0.556 **	−0.299 **	−0.012	0.032	0.116	0.131	0.331 **	−0.313 *	0.046	0.061	−0.040	−0.136	−0.030	0.665

** *p* < 0.01, * *p* < 0.05.

**Table 4 life-13-00424-t004:** Result of multiple stepwise regression analysis for independence of factors contributing to ONH MBR.

Dependent	Independent	*β*	*p*-Value
ONH MBR-vessel	CKD stage	−0.257	0.031
	Duration of DM	−0.185	0.115
	Total cholesterol	−0.165	0.163
	Ocular perfusion pressure	0.035	0.442
	Wall to lumen ratio	−0.070	0.586
	Triglyceride	0.060	0.609
	HbA1c	−0.038	0.752
	LDL	0.031	0.795
	Hb	0.016	0.892

**Table 5 life-13-00424-t005:** Result of multiple stepwise regression analysis for independence of factors contributing to TRFI.

Dependent	Independent	*β*	*p*-Value
TRFI	CKD stage	−0.316	0.015
	Duration of DM	−0.191	0.133
	Triglyceride	0.132	0.297
	HbA1c	−0.101	0.436
	Total cholesterol	−0.087	0.498
	Wall to lumen ratio	−0.070	0.586
	Ocular perfusion pressure	−0.041	0.747
	LDL	0.032	0.805
	Hb	−0.028	0.829

## Data Availability

The datasets generated during and/or analyzed during the current study are available from the corresponding author on reasonable request.

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
