# Peer review of "Relationship Between Retinal Microcirculation and Renal Function in Patients with Diabetes and Chronic Kidney Disease by Laser Speckle Flowgraphy"

_life, 2023, doi:10.3390/life13020424_

Round 1

Reviewer 1 Report

This is an interesting study about relationship between retinal microcirculation and renal function.

1. The authors' study is about not whole ocular circulation but only retina, so the title might be better to be changed into retinal microcirculation.   

2. Did you measure the WLR in only one area with adaptive optics imaging?

As authors know the AO can measure WLR in very small area,  if possible or needed, I recommend at least 2 area(superior and inferior) to provide more confidence of this study.

3. in line 212, the reference 35 is not proper, so the new reference for AO measurement should be needed. 

Author Response

  1. The authors' study is about not whole ocular circulation but only retina, so the title might be better to be changed into retinal microcirculation.

Answer: We agree with the reviewer and have revised the title.

  1. Did you measure the WLR in only one area with adaptive optics imaging? As authors know the AO can measure WLR in very small area, if possible or needed, I recommend at least 2 area (superior and inferior) to provide more confidence of this study.

Answer: We entirely agree with the reviewer. Although we confirmed the value by two retinal specialists, we measured the parameters measured by AO camera imaging such as the WLR at one site per eye, but we believe that more accurate results can be obtained by measuring at multiple sites per eye. We have described this in line 218 and limitation section in line 369.

  1. in line 212, the reference 35 is not proper, so the new reference for AO measurement should be needed.

Answer: We thank the reviewer for pointing it out. We have corrected the reference 35.

Reviewer 2 Report

The purpose of this study was to investigate the effect of renal dysfunction categorized with the stage of CKD (no, stage 1-2, stage 3) on the retinal microcirculation assessed by LSFG and retinal artery caliber measured by AO imaging in 115 patients with diabetes especially the early stage of retinopathy and nephropathy.  Retinal microcirculation (blood flow velocity) was decreased in response to renal dysfunction (stage 3), but there was no significant difference in the caliber of retinal vessels. These results suggest that impaired renal function may be associated with reduced retinal blood flow in early diabetic retinopathy. All limitations (3) are mentioned and argued well. Figures and Tables are very o.k., References up-to-date and sufficient. Congratulations to the authors, a perfectly designed and performed study with high clinical value

Very minor comments:

Maybe the authors could explain the abbreviations when used first (line 139 AL, line 168 RFV).

Please check the sentences lines 202-204 (some redundance)

Word hyphenation (line 279 dia-meter, line 304 dia-lysis)

Line 337: Feng instead of FENG

Author Response

The purpose of this study was to investigate the effect of renal dysfunction categorized with the stage of CKD (no, stage 1-2, stage 3) on the retinal microcirculation assessed by LSFG and retinal artery caliber measured by AO imaging in 115 patients with diabetes especially the early stage of retinopathy and nephropathy.  Retinal microcirculation (blood flow velocity) was decreased in response to renal dysfunction (stage 3), but there was no significant difference in the caliber of retinal vessels. These results suggest that impaired renal function may be associated with reduced retinal blood flow in early diabetic retinopathy. All limitations (3) are mentioned and argued well. Figures and Tables are very o.k., References up-to-date and sufficient. Congratulations to the authors, a perfectly designed and performed study with high clinical value.

Very minor comments:

  1. Maybe the authors could explain the abbreviations when used first (line 139 AL, line 168 RFV).

Answer: We have explained the abbreviations those.

  1. Please check the sentences lines 202-204 (some redundance)

Answer: We have modified the sentence.

  1. Word hyphenation (line 279 dia-meter, line 304 dia-lysis)

Answer: We thank the reviewer for your suggestion, but the hyphenation is due to this word system and could not be corrected.

  1. Line 337: Feng instead of FENG

Answer: We have corrected it.